# Axon-like protrusions promote small cell lung cancer migration and metastasis

**Dian Yang[1,2,3], Fangfei Qu[2,3], Hongchen Cai[3], Chen-Hua Chuang[3], Jing Shan Lim[1,2,3], Nadine Jahchan[2,3], Barbara M Grüner[3,4,5,6], Christin S Kuo[2], Christina Kong[4], Madeleine J Oudin[7], Monte M Winslow[1,3,4]\*, Julien Sage[1,2,3]\***

[1]Cancer Biology Program, Stanford University School of Medicine, Stanford, United States; [2]Department of Pediatrics, Stanford University School of Medicine, Stanford, United States; [3]Department of Genetics, Stanford University School of Medicine, Stanford, United States; [4]Department of Pathology, Stanford University School of Medicine, Stanford, United States; [5]Department of Medical Oncology, West German Cancer Center, University Hospital Essen, Essen, Germany; [6]German Cancer Consortium (DKTK) partner site Essen, Essen, Germany; [7]Department of Biomedical Engineering, Tufts University, Medford, United States

**Abstract** Metastasis is the main cause of death in cancer patients but remains a poorly understood process. Small cell lung cancer (SCLC) is one of the most lethal and most metastatic cancer types. SCLC cells normally express neuroendocrine and neuronal gene programs but accumulating evidence indicates that these cancer cells become relatively more neuronal and less neuroendocrine as they gain the ability to metastasize. Here we show that mouse and human SCLC cells in culture and in vivo can grow cellular protrusions that resemble axons. The formation of these protrusions is controlled by multiple neuronal factors implicated in axonogenesis, axon guidance, and neuroblast migration. Disruption of these axon-like protrusions impairs cell migration in culture and inhibits metastatic ability in vivo. The co-option of developmental neuronal programs is a novel molecular and cellular mechanism that contributes to the high metastatic ability of SCLC.

**\*For correspondence:**
mwinslow@stanford.edu (MMW);
julsage@stanford.edu (JS)

**Competing interest:** See
page 15

**Reviewing editor:** Jonathan A
Cooper, Fred Hutchinson Cancer
Research Center, United States

## Introduction

Metastases are a major cause of cancer-related morbidity and mortality. By the time cancer cells leave their primary site and spread to distant sites, they have acquired the ability to migrate and invade, as well as characteristics that enable them to survive and proliferate within new microenvironments. These phenotypes are likely driven by changes in gene expression and epigenetic programs that allow cancer cells to overcome the many hurdles that normally constrain the metastatic process. Despite recent advances, our understanding of the principles and mechanisms underlying metastasis remains incomplete, including how changes in molecular programs can translate into selective advantages that enable cancer cells to spread to other organs (*Fidler, 2003*; *Obenauf and Massagué, 2015*; *Lambert et al., 2017*).

Small cell lung carcinoma (SCLC) is a high-grade neuroendocrine cancer that accounts for ~15% of all lung cancers and causes over 200,000 deaths worldwide each year (*Sabari et al., 2017*). The ability of SCLC cells to leave the primary tumor and establish inoperable metastases is a major cause of death and a serious impediment to successful therapy (*van Meerbeeck et al., 2011*; *Farago and Keane, 2018*). SCLC is one of the most metastatic human cancers, with over 60% of SCLC patients presenting with disseminated disease at the time of diagnosis, often including liver, bone, brain, and secondary lung metastases (*Nakazawa et al., 2012*; *Riihimäki et al., 2014*).

Molecular analyses to understand metastatic progression of human cancer are often limited by difficulties in accessing tumor samples at defined stages. This problem is especially true for SCLC,

since patients with metastatic disease rarely undergo surgery (*Barnes et al., 2017*). Genetically engineered mouse models of human SCLC recapitulate the genetics, histology, therapeutic response, and highly metastatic nature of the human disease (*Kwon and Berns, 2013*; *Gazdar et al., 2015*; *Rudin et al., 2019*). These genetically engineered mouse models recapitulate cancer progression in a controlled manner and allow for the isolation of primary tumors and metastases directly from their native microenvironment. Recently, we and others have used mouse models to uncover gene expression programs that are altered in SCLC metastases (*Denny et al., 2016*; *Semenova et al., 2016*; *Wu et al., 2016*; *Yang et al., 2018*). While SCLC cells display features of neuroendocrine cells, the gene expression programs in metastatic SCLC include not only genes normally expressed in pulmonary neuroendocrine cells but also those expressed in neurons (*Carney et al., 1982*; *Cutz, 1982*; *Broers et al., 1987*; *Anderson et al., 1988*). Higher levels of the neuronal markers such as NSE (neuron-specific enolase) correlate with shorter survival and more metastatic disease in SCLC patients (*Carney et al., 1982*; *van Zandwijk et al., 1992*; *Dong et al., 2019*). Broad neuronal gene expression programs are enriched in metastases from mouse models of SCLC, however, whether SCLC cells actually gain neuronal characteristics and whether neuronal features are key regulators of metastatic ability has not been previously characterized (*Denny et al., 2016*; *Wu et al., 2016*; *Yang et al., 2018*; *Böttger et al., 2019*).

Here we find that the metastatic state of SCLC can include the growth of protrusions that resemble axons. These axon-like growths increase the ability of SCLC cells to migrate and metastasize, thus representing a cellular mechanism that enhances the metastatic ability of SCLC cells that have transitioned to a more neuronal cell state.

## Results

### SCLC cells can form long cellular protrusions in culture and in vivo

To investigate SCLC migration, we developed an assay in which SCLC cells, which classically grow in culture as floating spheres or aggregates, are grown as a monolayer under Matrigel (*Denny et al., 2016* and Materials and methods). Unexpectedly, we noticed that cells from some SCLC cell lines (N2N1G, 16T, 6PF) derived from the $Rb^{f/f};Trp53^{f/f}$ (*DKO*) and $Rb^{f/f};Trp53^{f/f};p130^{f/f}$ (*TKO*) genetically engineered mouse models form long cellular protrusions into cell-free spaces (*Figure 1A–B*). To determine whether these structures specifically project into cell-free areas or they also exist within monolayers, we cultured a minor fraction of fluorescently-labeled, GFP$^{positive}$ SCLC cells with control SCLC cells. We found that SCLC cells also form protrusions when they are in close contact with surrounding cancer cells (*Figure 1—figure supplement 1A*). Similar mixing experiments performed in subcutaneous allografts also documented the growth of protrusions by SCLC cells in vivo (*Figure 1C–D*). Finally, similar structures also extend from SCLC micro-metastases in the liver in the autochthonous *TKO* mouse model and after intravenous transplantations of SCLC cells (*Figure 1—figure supplement 1B–C*).

Human SCLC patient-derived xenografts (PDXs) recapitulate many important features of the human disease (e.g. *Saunders et al., 2015*; *Gardner et al., 2017*). To label rare cancer cells within human SCLC PDXs and identify whether they had protrusions in unperturbed tumors, we used DiI tracing. DiI is a lipophilic dye that diffuses within cell membranes and has been widely employed to label projections from individual neurons (*Mufson et al., 1990*; *Heilingoetter and Jensen, 2016*). Protrusions from SCLC cells were easily identifiable in two out of three PDX models (*Figure 1E–F*). Furthermore, human NCI-H446 SCLC cells formed long protrusions into cell-free areas in the 2D culture system (*Figure 1—figure supplement 1D–E*) and when grown as xenografts (*Figure 1—figure supplement 1F*). Other human and mouse cells had a variable capacity to form protrusions (*Figure 1—figure supplement 1G–H*, *Supplementary file 1*: Key Resources table, and *Supplementary file 2*–table 1).

These observations indicated that at least a subset of SCLC cells, which are often described as being 'small round blue' cells, can develop long cellular protrusions. We next sought to investigate the nature of these protrusions and uncover their possible role in metastatic SCLC.

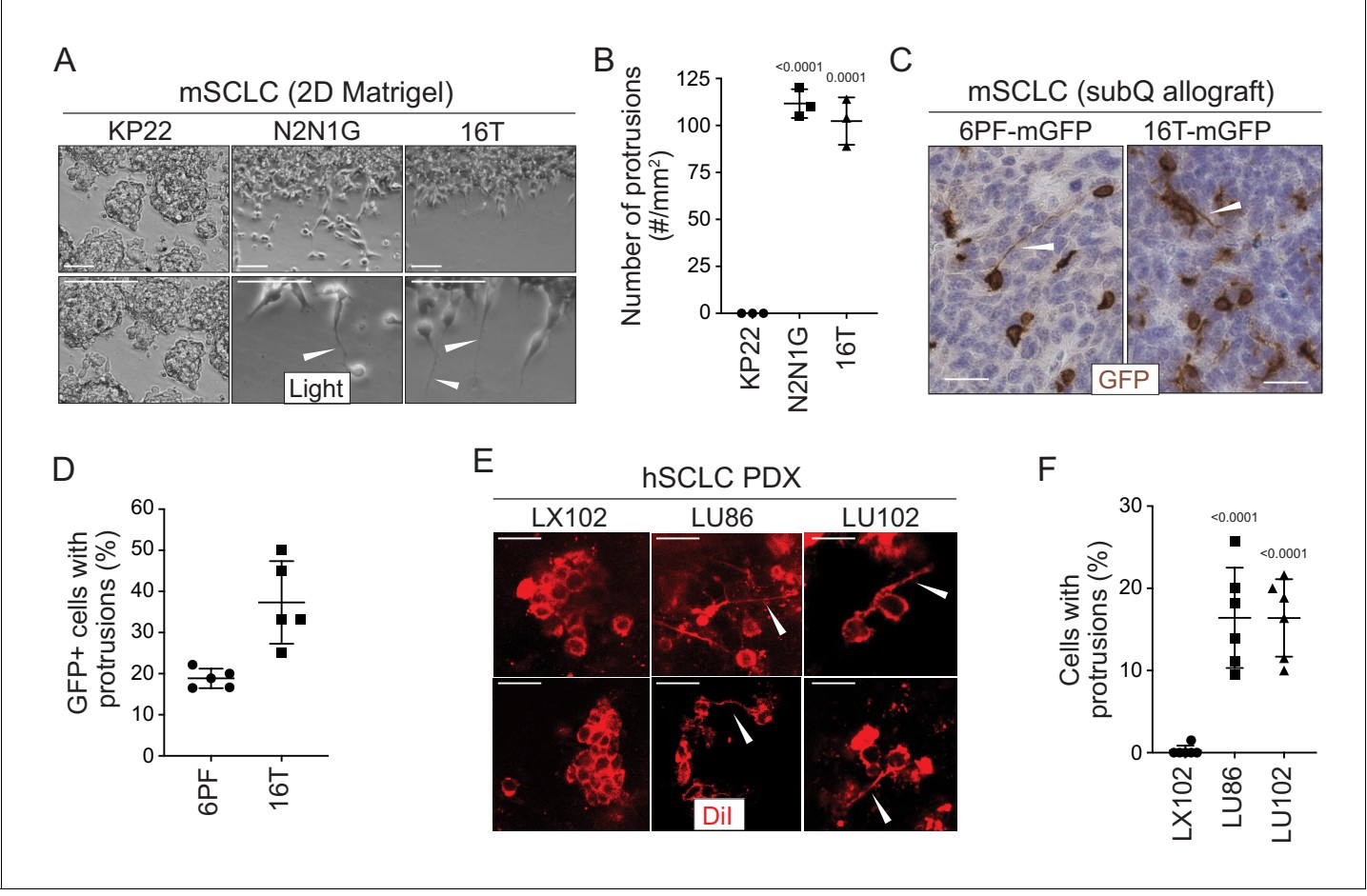

**Figure 1.** SCLC cells grow protrusions in culture and in vivo. (**A**) Representative bright field images of three murine SCLC (mSCLC) cell lines (KP22, N2N1G, and 16T). Cells extend protrusions into a cell-free scratch generated in monolayer cultures. Protrusions are indicated by white arrowheads. Scale bars, 100 μm. N = 3 replicates. (**B**) Quantification of the number of protrusions that form from each mSCLC cell line as cultured in (**A**). Each symbol corresponds to the average of two technical replicates of an independent experiment. Mean + /- s.d. is shown, unpaired t-test. (**C**) Representative images of mSCLC cells (6PF and 16T) growing as subcutaneous tumors. At the time of injection, 10% SCLC cells stably expressing membrane-GFP (mGFP) were mixed with 90% GFP-negative SCLC cells. Immunostaining for GFP generates a brown signal. Examples of protrusions are indicated by white arrowheads. Hematoxylin (blue) stains the nuclei of the cells. (N = 5/allograft, from one biological replicate). Scale bar, 20 μm. (**D**) Quantification of (**C**). Each symbol represents an allograft tumor (N = 4/allograft, from one biological replicate). Mean + /- s.d. is shown. (**E**) Representative images of human SCLC (hSCLC) patient-derived xenografts growing subcutaneously (LX102, LU86, and LU102 models). Tumors were injected with the red fluorescent tracer DiI. Protrusions are indicated by white arrowheads. Scale bar, 20 μm. (**F**) Quantification of (**E**). Each symbol represents a xenograft tumor (N = 6/xenograft, from one biological replicate). Mean + /- s.d. is shown, unpaired t-test.
The online version of this article includes the following figure supplement(s) for figure 1:

**Figure supplement 1.** SCLC cells grow protrusions in culture and in vivo.

## SCLC protrusions resemble axons and SCLC cells with protrusions migrate similar to neuroblasts

SCLC cells express typical neuroendocrine genes but also neural and neuronal genes (*Carney et al., 1982*; *Cutz, 1982*). This observation led us to investigate whether the protrusions were similar to neuronal axons or dendrites. We identified a list of 69 genes classically associated in the scientific literature with axonogenesis and axon guidance, and found that many of these genes are expressed in at least subsets of primary human SCLCs (*George et al., 2015*) (*Supplementary file 2*–table 2). Thus, the gene expression programs controlling axonal growth in neuronal cells are also present in SCLC cells. We previously performed gene expression analyses on purified cancer cells from primary tumors and metastases from two mouse models of SCLC (*Denny et al., 2016*; *Yang et al., 2018*). In these studies, we found a general increase in the expression of neuronal gene expression programs

during tumor progression, with broad expression of the selected candidate genes in metastatic SCLCs, indicating that murine SCLC tumors and cell lines derived from these tumors represent a tractable system with which to investigate neuronal programs in SCLC (*Supplementary file 2*–table 3). Pathway and process enrichment analysis on these 69 genes confirmed their connection with axon guidance, neuron migration, and nervous system development (*Supplementary file 2*–table 4).

To further investigate the nature of these SCLC protrusions, we assessed their expression of canonical axonal and dendritic proteins. The protrusions that form from murine and human SCLC cell lines uniformly expressed neuron-specific class III beta-tubulin (Tuj1). More importantly, these protrusions were positive for the axonal marker TAU while expression of the dendritic marker MAP2 was undetectable (*Figure 2A–B* and *Figure 2—figure supplement 1A–C*). Tuj1$^{positive}$, TAU$^{positive}$ protrusions were also observed in vivo emanating from SCLC cells in the liver of *TKO* mice (*Figure 2—figure supplement 2A*). Furthermore, 29/79 (37%) human primary SCLC tumors stained moderately or strongly positive for TAU (*Figure 2—figure supplement 2B*). Most axonogenesis and neuronal migration genes were undetectable in a single-cell RNA-seq analysis of adult lung epithelial cells, which included neuroendocrine cell, further suggesting that these programs are turned on during tumorigenesis (*Figure 2—figure supplement 2C*) (*Ouadah et al., 2019*). Immunostaining for the axonal marker GAP43 (which is highly expressed in the metastatic SCLC state; see below) did not uncover any positive normal lung epithelial cells (*Figure 2—figure supplement 2D*). Together, these observations indicate that axonal programs are gained during SCLC progression and suggested that the protrusions from SCLC cells are axon-like.

We quantified the length of protrusions and found that they were often 5 to 10 times longer than the diameter of the cell body (~8 μm) (*Figure 2C*). The length and the frequency of these axon-like protrusions suggested that they might influence the behavior of SCLC cells. We investigated and quantified the features of SCLC cells with and without protrusions using time-lapse microscopy. Initial observations of mouse SCLC cells showed that the protrusions were very dynamic (*Figure 2D* and *Video 1*). In these videos, we noticed that the protrusions resembled cellular processes that have been described in neuroblasts and with the movement of SCLC cells along these protrusions reminiscent of neuronal tangential migration exhibited by neuroblasts (*Lois et al., 1996*; *Oudin et al., 2011*; *Zhou et al., 2015*) and interneurons (*Leclech et al., 2019*). Indeed, when we quantified the movement of SCLC cell along protrusions, SCLC cell lines that form protrusions (16T and N2N1G cell lines) displayed increased saltatory activity compared to SCLC cells that do not form protrusions (KP22 cell line) (*Figure 2D–H* and *Video 2*, *3*, and *4*). The velocity of SCLC cells that form protrusions was also greatly increased compared to cells that do not form protrusions (*Figure 2—figure supplement 1D*).

Together, these results indicate that SCLC cells can generate axon-like protrusions and that these projections facilitate migration in a manner that is qualitatively similar to neuroblast migration during brain development.

## Expression of a gene signature for axonogenesis and neuronal migration across SCLC subtypes

To investigate the functional importance of these axon-like protrusions and their regulation, we focused on 13 genes (out of the 69 genes selected above) that encode for proteins functionally involved in diverse aspects of axon formation, axon guidance, and neuronal migration (*Supplementary file 2*–table 5). These 13 genes are all expressed in at least a subset of human SCLC tumors (*Figure 3—figure supplement 1A*) (data from *George et al., 2015*). We excluded gene families for which functional overlap and compensatory mechanism were likely. STRING analysis and literature searches confirmed that these 13 candidates had a significant connection with biological processes related to neurogenesis and the regulation of neuron projection development. These proteins were not often connected with one another and thus likely contribute to distinct aspects of these biological processes (*Figure 3—figure supplement 1B* and *Supplementary file 2*–table 6).

A better understanding of the mechanisms that lead to the upregulation of gene programs linked to axonogenesis and neuronal migration may help us understand the functional role of these gene programs in SCLC cells. SCLC tumors have been divided in major subtypes driven by key transcription factors (*Rudin et al., 2019*). In human tumors (*George et al., 2015*), the 13-gene signature correlated more closely with the 'SCLC-N' subtype, driven by the transcription factor NEUROD1, and

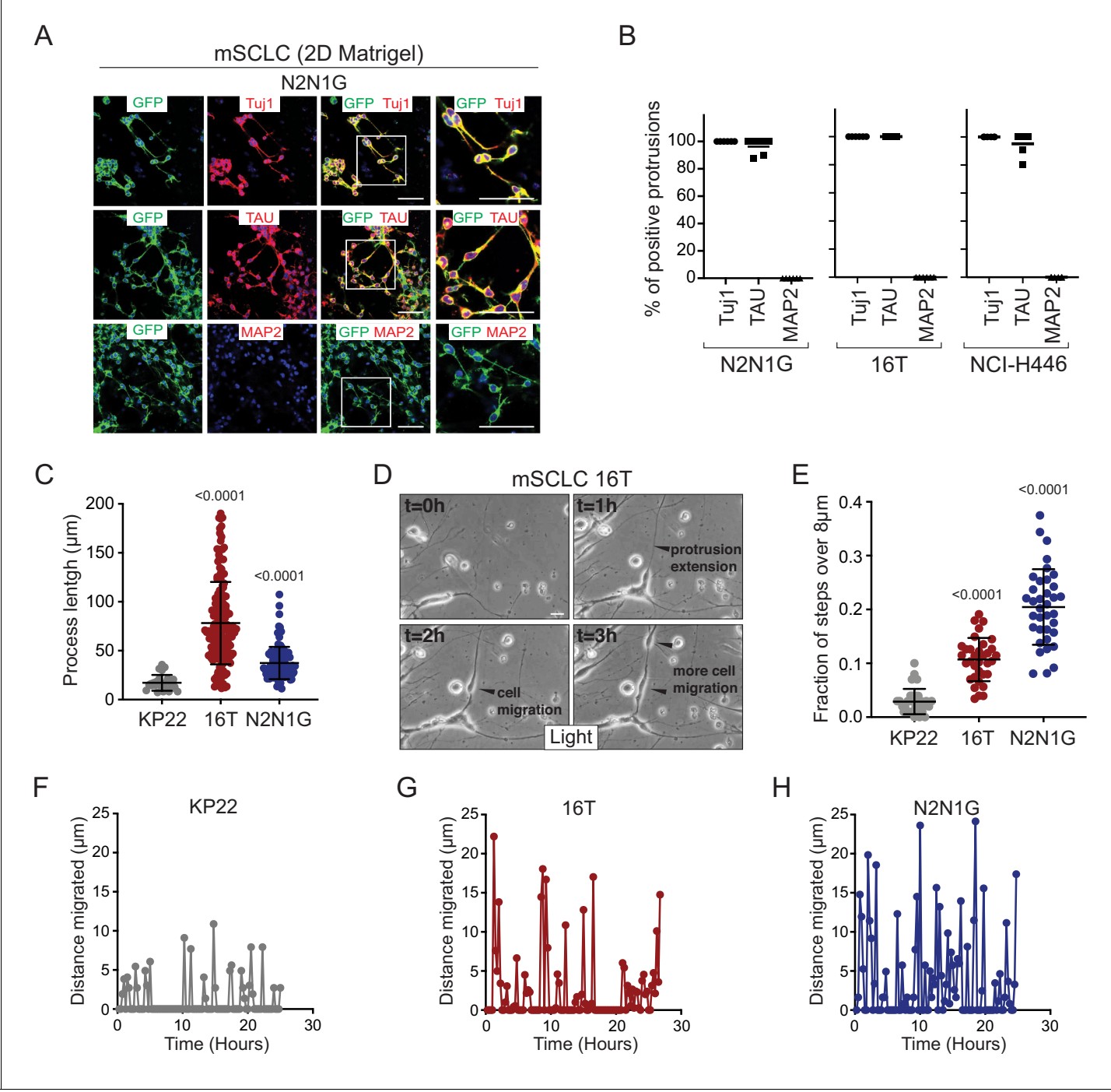

**Figure 2.** SCLC cells with protrusions migrate in a saltatory fashion similar to neuroblasts. (**A**) Representative immunofluorescence images of N2N1G mSCLC cells expressing membrane-GFP (GFP, green) and stained (red) for expression of the neuronal marker Tuj1, the axonal marker TAU, or the dentritic marker MAP2. DAPI marks the nucleus of cells in blue. Scale bars, 50 μm. (**B**) Quantification of (**A**) for two mouse SCLC cell lines (16T, N2N1G) and one human SCLC cell line (NCI-H446). Images for 16T and NCI-H446 are shown in *Figure 2—figure supplement 1B–C*. N = 5/cell line. The bar is the mean. (**C**) Quantification of the length of protrusions in three mSCLC cell lines (KP22, no visible protrusions, 16T and N2N1G with protrusions). The average cell size in these experiments was ~8 μm. Each dot represents a cell. N > 10 fields were quantified in one biological replicate. Mean + / - s.d. is shown, Mann-Whitney test. (**D**) Representative still images from time-lapse videomicroscopy analysis of 16T SCLC cells showing the dynamic nature of the protrusions (from *Video 1*). (**E**) Quantification of the saltatory movements of three mSCLC cell lines as indicated. Note the correlation between the presence of protrusions and the ability of making longer steps (longer than the average cell size). Each dot represents a cell. N > 10 fields were

*Figure 2 continued on next page*

*Figure 2 continued*

quantified in one biological replicate. Mean + /- s.d. is shown, Mann-Whitney test. (**F–H**) Example of single cell movement over time for each of the three mSCLC cell lines.

The online version of this article includes the following figure supplement(s) for figure 2:

**Figure supplement 1.** SCLC protrusions resemble axons and enable rapid cell movement.
**Figure supplement 2.** Mouse and human SCLC cells express axonal markers in vivo.

both human cell line NCI-H446 and the PDX model LU86 (*Saunders et al., 2015*) belong to this subtype (*Figure 3—figure supplement 1C*). The murine cell lines used to study protrusions in this study are of the 'SCLC-A' subtype (driven by the transcription factor ASCL1), even though the correlation between the 13-gene signature and ASCL1 expression was weak in mouse tumors (*Yang et al., 2018*) (*Figure 3—figure supplement 1C*). We also found no correlation between *ASCL1* and *NEUROD1* expression and the ability to grow protrusions in other human cell lines (*Figure 3—figure supplement 1D* and *Supplementary file 2*–table 1). Thus, the ability to grow protrusions may exist across SCLC subtypes. We and others have identified a role for the NFIB transcription factor in SCLC metastasis and the induction of gene programs linked with neuronal differentiation (*Denny et al., 2016*; *Semenova et al., 2016*; *Wu et al., 2016*). Notably, the 13-gene signature correlated with high NFIB expression (*Figure 3—figure supplement 1C*). NFIB knock-down in mouse SCLC cells that had high NFIB levels and formed protrusions did not result in the global downregulation of the 13-gene signature, but was sufficient to reduce the formation of protrusions (*Figure 3—figure supplement 1E–F*). Overexpression of NFIB in SCLC mouse cells with low NFIB levels and without protrusions was not sufficient to lead to the upregulation of the entire set of 13 genes or to induce the growth of protrusions (*Figure 3—figure supplement 1E,G*). Thus, while NFIB upregulation may be important in the induction of neuronal programs in SCLC cells, the upstream factors that control neuronal programs specifically associated with axonogenesis and migration in SCLC remain to be fully characterized. These experiments led us to more specifically test the role of the 13 selected genes in the formation of protrusions and the role of these protrusions in cell migration and metastasis.

## Loss of Axon-like protrusions inhibits the migration of SCLC cells

In the 25 human SCLC cell lines analyzed in the Cancer Dependency Map project, knock-down of these 13 genes rarely affected the growth of SCLC cells in culture, consistent with these genes influencing aspects of cell physiology not related to the cell cycle (*Supplementary file 2*–table 7 and *Figure 3—figure supplement 2A*). We performed immunostaining for one of these 13 proteins (GAP43) and found that ~50% of human primary SCLC tumors stained moderately or strongly positive (*Figure 3—figure supplement 2B*), further supporting a role for neuronal programs linked to axonogenesis and migration in SCLC.

The 13-gene signature was overall more highly expressed in N2N1G cells, which are derived from a lymph node metastasis and grow protrusions compared to KP22 cells that do not grow protrusions (*Figure 3—figure supplement 2C*). We first knocked-down each of these 13 genes with two shRNAs in N2N1G cells. We confirmed stable knock-down by RT-qPCR (*Supplementary file 2*–table 8) and quantified the development of protrusions in the monolayer culture assay. Knock-down of 11 of the 13 genes significantly reduced the number of protrusions with at least one shRNA (*Figure 3A–B* and *Figure 3—figure supplement 3A*). Knock-down of multiple factors normally implicated in distinct steps of axonal growth reduced the development of protrusions from SCLC cells, thus further

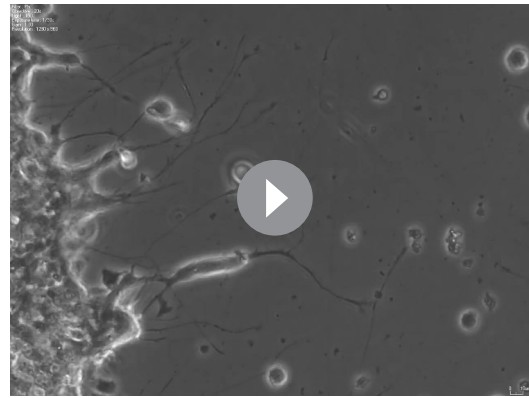

**Video 1.** Time-lapse video of 16T mouse SCLC cells (images collected every 15 min).
https://elifesciences.org/articles/50616#video1

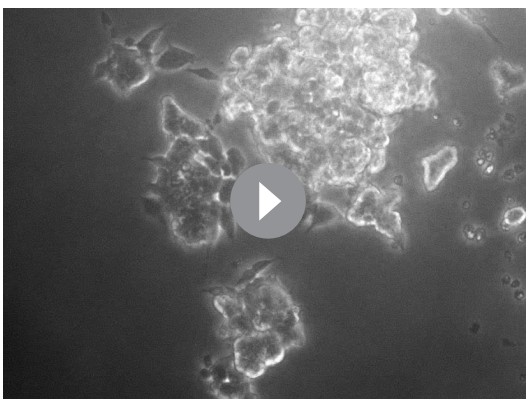

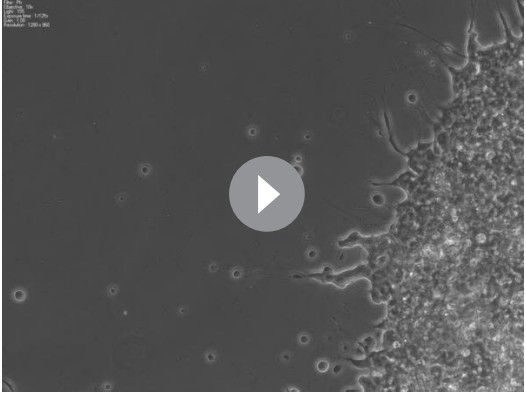

**Video 2.** Time-lapse video of KP22 mouse SCLC cells (images collected every 15 min).
https://elifesciences.org/articles/50616#video2

**Video 3.** Time-lapse video of 16T mouse SCLC cells (images collected every 15 min).
https://elifesciences.org/articles/50616#video3

bolstering the notion that these protrusions are similar to neuronal axons. Knock-down of the many genes involved in axon formation, axonal guidance, and neuronal migration also reduced cell migration (*Figure 3B*). Quantification of cell migration showed that inhibition of migration correlated with loss of the axon-like protrusions (*Figure 3C–D*). We validated the knock-down for two of the top candidates, *Gap43* and *Fez1* genes, by immunoblot for the corresponding proteins in N2N1G cells (*Figure 3—figure supplement 3B–C*). We further validated the effects of knocking down these two factors on the growth of protrusions and cell migration in a second SCLC cell line (16T; *Figure 3E–J* and *Figure 3—figure supplement 3D–E*). Finally, we found that knock-down of *Gap43* and *Fez1* reduced the ability of SCLC 16T and N2N1G cells to migrate out of 3D spheroids in Matrigel (*Figure 3—figure supplement 3F–G*).

Together, these data show that SCLC cells with axon-like protrusions migrate in culture similar to what has been described for neuroblasts and that disruption of these protrusions by knocking down a variety of diverse genes involved in axonogenesis and neuronal migration also reduces SCLC migration.

## Knock-down of genes associated with the formation of protrusions decreases metastatic potential

The link between axon-like protrusions and migration in vitro led us to investigate whether these axon-like protrusions promote the metastatic ability of SCLC cells in vivo. In support of this idea, we found that the expression of neuron-specific class III beta-tubulin and TAU was barely detectable in non-metastatic tumors in the lungs of *TKO* mice 3 months after cancer initiation while a majority of later stage tumors stained strongly positive for both proteins (*Figure 4—figure supplement 1A–B*). GAP43 was detectable in 6/9 human SCLC metastases analyzed (*Figure 4—figure supplement 1C*). We also found a significant increase of key genes involved in axonogenesis and neuronal migration in metastases compared to primary tumors in a mouse model of SCLC (*Figure 4—figure supplement 1D*).

To test the role of these protrusions in the metastatic process in vivo, we investigated whether SCLC cells with *Gap43* or *Fez1* knocked-down had reduced metastatic ability. The products of these genes are thought to regulate axonal development in distinct manners but knock-down of each reduced the formation of protrusions and cell migration in culture. We first assessed whether *Gap43* and Fez1 knock-

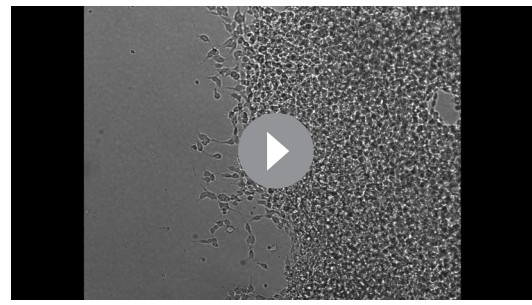

**Video 4.** Time-lapse video of N2N1G mouse SCLC cells (images collected every 15 min).
https://elifesciences.org/articles/50616#video4

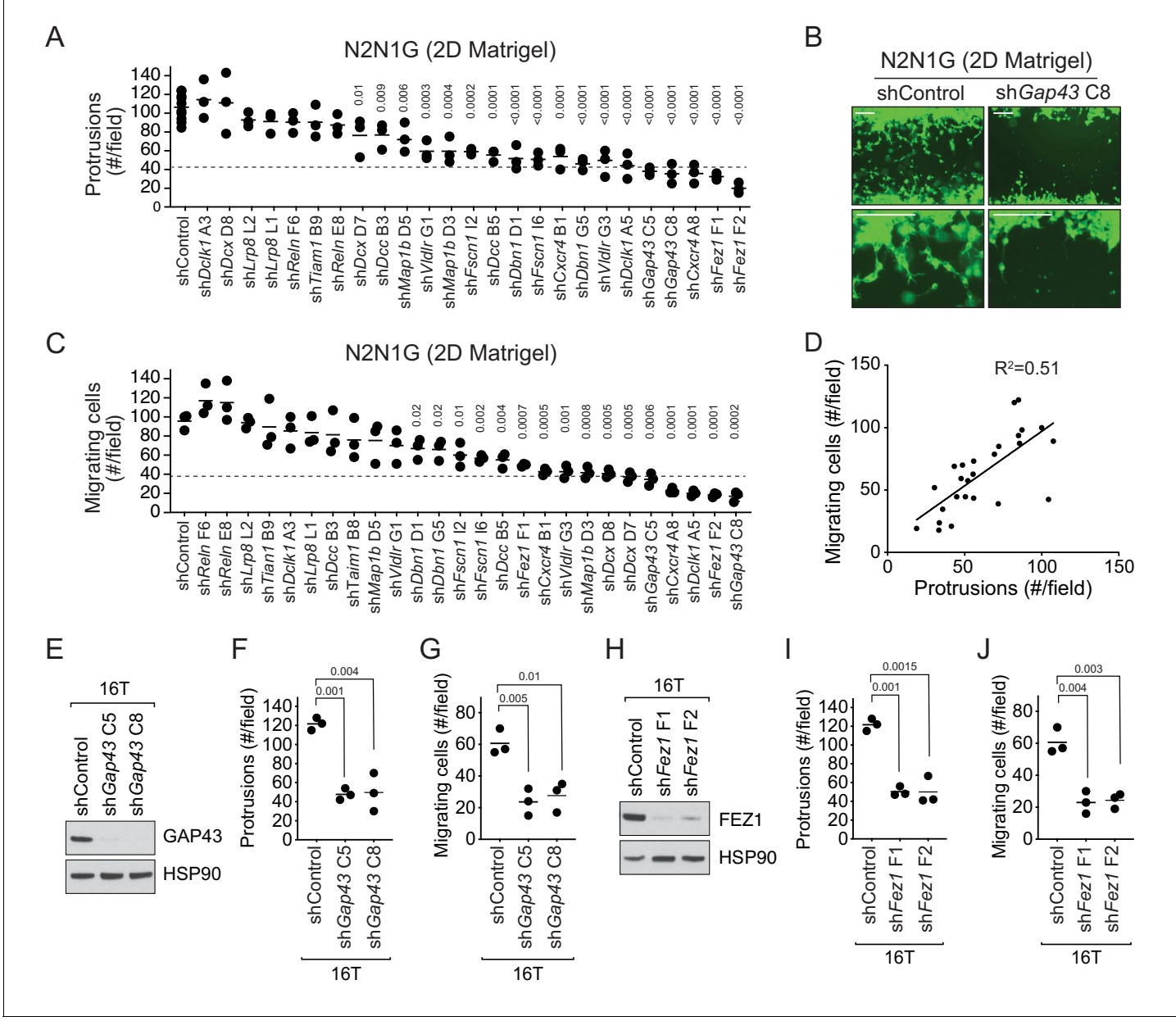

**Figure 3.** The axonal-like protrusions contribute to the migratory ability of SCLC cells in culture. (**A**) Quantification of the number of cells with protrusions when mGFP-labeled N2N1G mSCLC cells were allowed to grow into a cell-free scratch generated in monolayer cultures under Matrigel. N = 3 independent experiments (shControl, N = 3 per experiment, total N = 9 plotted together). An unpaired t-test was used for statistical analysis and p-values are shown. Only significant p-values are shown. The dotted line represents a 60% reduction compared to the mean value of the controls. (**B**) Representative images of the data quantified in (**A**) and (**C**) with knock-down of *Gap43*. Scale bars, 100 μm. (**C**) Quantification of the migration of cells with protrusions when mGFP-labeled N2N1G mSCLC cells were allowed to grow into a cell-free scratch generated in monolayer cultures under Matrigel. N = 3 independent experiments. An unpaired t-test was used for statistical analysis and p-values are shown. Only significant p-values are shown. The dotted line represents a 60% reduction compared to the mean value of the controls. (**D**) Correlation of the data in (**A**) and (**C**) using the mean value for each knock-down. Pearson correlation $R^2$ value is shown. (**E** and **H**) Immunoblot analysis of GAP43 or FEZ1 levels, respectively, in control and knock-down 16T mSCLC cells. HSP90 is a loading control. (**F** and **I**) Quantification of the number of cells with protrusions as in (**A**) with 16T mSCLC cells and *Gap43* or *Fez1* knock-down, respectively (N = 3). An unpaired t-test was used for statistical analysis and p-values are shown. (**G** and **J**) Quantification of the migration of cells with protrusions as in (**B**) with 16T mSCLC cells and *Gap43* or *Fez1* knock-down, respectively (N = 3). An unpaired t-test was used for statistical analysis and p-values are shown.

The online version of this article includes the following figure supplement(s) for figure 3:

**Figure supplement 1.** The expression of the 13 genes selected for their possible role in the formation of protrusions is in part regulated by NFIB.

*Figure 3 continued on next page*

*Figure 3 continued*

**Figure supplement 2.** The 13 genes selected for their possible role in the formation of protrusions are expressed in human SCLC but do not play a key role in the expansion of SCLC cell populations.

**Figure supplement 3.** Knock-down of GAP43 and FEZ1 disrupts the formation of protrusions and cell migration in mouse SCLC cell lines in culture.

down reduced the metastatic ability of mouse N2N1G SCLC cells after intravenous transplantation of control and knock-down cells into recipient mice. Knock-down of each of these pro-protrusion factors significantly reduced the number of metastases as assessed by tumor counts at the surface of the liver 4–5 weeks after intravenous injection (*Figure 4—figure supplement 2A–B*). To determine whether GAP43 and FEZ1 are simply required for tumor growth in vivo, we transplanted *Gap43* and *Fez1* knock-down cells subcutaneously and quantified tumor growth. Knock-down of these genes had no effect on subcutaneous tumor growth suggesting that the effects on metastatic ability likely represent the disruption of phenotypes uniquely associated with the metastatic process (*Figure 4—figure supplement 2C*). We repeated these experiments with two independent shRNAs for each gene in both N2N1G and 16T SCLC cells, which confirmed that *Gap43* and *Fez1* knock-down reduced the formation of liver metastases after intravenous injection of SCLC cells (*Figure 4A–H* and *Figure 4—figure supplement 2D–E*).

The absence of growth defects in subcutaneous tumors following *Gap43* and *Fez1* knock-down suggested that these genes may affect earlier steps of the metastatic cascade. To test this, we performed similar intravenous transplant experiments but quantified the presence of SCLC cells in the liver 2 days after injection (*Figure 4I*). Quantification of GFP$^{positive}$ cancer cells in the liver by flow cytometry documented a significant reduction in metastatic seeding by SCLC cells with *Gap43* or *Fez1* knocked-down (*Figure 4J–M* and *Figure 4—figure supplement 2F–I*). Thus, reduced expression of genes associated with the formation of axon-like protrusions affects early metastatic seeding of SCLC cells in the liver, which ultimately translates to reduced metastatic burden.

## Discussion

While metastasis remains a major cause of morbidity and mortality in SCLC patients, its underlying mechanisms remain poorly understood and no therapeutic strategies exist to prevent metastatic spread or specifically treat metastatic SCLC. Here we investigated the function of neuronal gene expression programs in metastatic SCLC. We found that SCLC cells can grow axon-like protrusions and that these protrusions contribute to the migratory and metastatic phenotypes of these cells. This study identifies a cellular mechanism by which a neuroendocrine-to-neuronal transition promotes metastasis of SCLC cells.

The expression of neuronal factors in SCLC has been known for more than three decades and has been used as a marker for disease progression (*Carney et al., 1982*; *Cutz, 1982*; *Broers et al., 1987*; *Anderson et al., 1988*). However, whether neuronal programs in SCLC cells play a direct role in SCLC progression has not been rigorously investigated. We uncovered the formation of axon-like protrusions as one functional aspect of neuronal differentiation in SCLC and provide data to support a role for these protrusions in migration and metastasis. It is likely that other phenotypes usually associated with neurons beyond these axon-like protrusions also contribute to the expansion and the spread of SCLC cells. Furthermore, these axon-like protrusions may have other functions beyond facilitating metastatic seeding to the liver, which may including facilitating SCLC cell migration within the primary tumor, intravasation into the bloodstream, and movement within the parenchyma during metastatic expansion (*Shibue et al., 2012*). Future investigation of the roles of axon-like protrusions in SCLC will likely benefit from additional genetic analyses as well as high-resolution in vivo imaging methods. Recent evidence suggests that several other human tumor types also increase the expression of neuronal programs as they become more metastatic, especially to the brain (*Wingrove et al., 2019*). It will be important for future studies to determine if aspects of the neuronal program also contribute to the striking ability of SCLC cells to seed and expand in the brain (*Lukas et al., 2017*).

Our data indicate that SCLC metastasis is facilitated by the development of axon-like protrusions, however other molecular mechanisms certainly also increase the probability that a cancer cell will successfully overcome all the hurdles that limit the development of tissue destructive metastases.

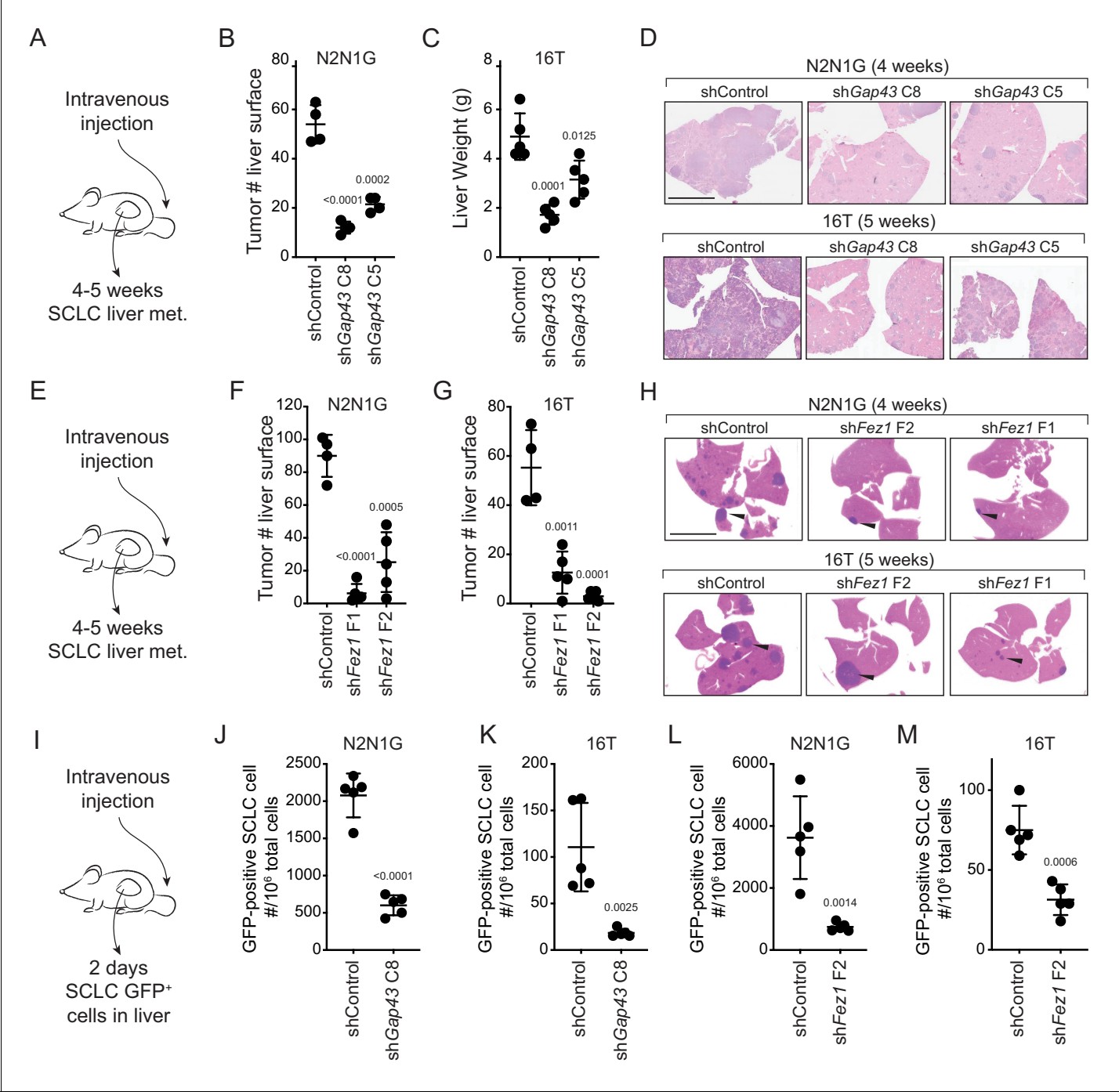

**Figure 4.** Genes involved in the generation of protrusions also control the formation of metastases. (**A**) Diagram of the approach to investigate the formation of liver metastases (met.) after intravenous injection of SCLC cells. (**B–C**) Quantification of the number of metastases 4 and 5 weeks after intravenous injection of N2N1G and 16T mSCLC cells, respectively, with control knock-down or knock-down of *Gap43* with two independent shRNAs. For N2N1G, tumors at the surface of the liver were quantified on the liver surface, as shown in *Figure 4—figure supplement 2D*. Too many tumors were present with the 16 T cell line and the control shRNA, and quantification was thus performed by measuring liver weight. N = 4–5 mice per condition in one biological replicate. Mean + /- s.d. unpaired t-test. (**D**) Representative hematoxylin and eosin (H and E) images of liver sections of mice in (**B–C**). Scale bars, 5 mm. (**E–H**) As shown in (**A–D**) for *Fez1* knock-down. See *Figure 4—figure supplement 2E* for representative images with N2N1G cells for the quantification in (**F–G**) of tumors at the surface of the liver. Arrows point to metastases. N = 4–5 mice per condition in one biological replicate. Mean + /- s.d. is shown, unpaired t-test. (**I**) Diagram of the approach to investigate early steps in liver metastasis, 2 days after intravenous injection. (**J–M**) Quantification of the number of GFP^positive N2N1G and 16T mSCLC cells 2 days after intravenous injection. See *Figure 4—figure supplement 2F*-ID for representative flow cytometry. N = 5 mice per condition in one biological replicate. Mean + /- s.d., unpaired t-test.

*Figure 4 continued on next page*

*Figure 4 continued*

The online version of this article includes the following figure supplement(s) for figure 4:

**Figure supplement 1.** Increased expression of axonal markers in metastatic SCLC.
**Figure supplement 2.** Reduced formation of metastasis upon knock-down of GAP43 and FEZ1 in SCLC cells.

For instance, we found that knock-down of *Dcx* (coding for Doublecortin) has little to no effect on the number of protrusions but strongly inhibits migration in our 2D Matrigel assay (*Figure 3A–B*), thus Doublecortin may promote SCLC migration independent from an impact on protrusion formation.

The formation of protrusions in SCLC cells is controlled by pathways previously implicated in the formation of axons and the migration of neuronal cells but it is unclear how the expression of these pro-protrusion genes is coordinated. Existing data support a role for the NFIB transcription factor across SCLC subtypes in the up-regulation of neuronal gene programs in general and gene programs associated more specifically with axonogenesis and neuronal migration (this study and *Denny et al., 2016*; *Semenova et al., 2016*; *Wu et al., 2016*). However, it is likely that a combination of genetic and epigenetic factors contributes to the ability of SCLC to grow protrusions (*Qadeer et al., 2019*). Adhesion molecules and other factors in the tumor microenvironment are also likely to contribute to the formation of protrusions in vivo (*Guo et al., 2000*).

Could an understanding of the molecular and cellular processes related to axon-like protrusions in SCLC cells ultimately be translated into clinical benefit for SCLC patients? Because NFIB is an important regulator of neuronal gene programs in SCLC cells, targeting this transcription factor may help inhibit SCLC metastatic potential in the future; one possible strategy could be the use of targeted proteolysis (*Paiva and Crews, 2019*). Another option could be to target individual factors in the axonogenesis and neuronal migration programs. Several of these factors may be required to drive these programs and these programs may not be as critical for brain function in adults as they are during development. Previous studies on SCLC have targeted the CXCR4 chemokine receptor due to its role in cell adhesion and migration and its expression in SCLC cells (*Burger et al., 2003*; *Teicher, 2014*; *Taromi et al., 2016*). Interestingly, CXCR4 also contributes to the formation of axon-like protrusions (*Figure 3*). In a recent clinical trial in SCLC patients, CXCR4 inhibition was well tolerated but this inhibition did not significantly reduce disease progression (*Salgia et al., 2017*). However, agents that reduce the ability of cancer cells to overcome early barriers of metastatic seeding will likely need to be employed in specific settings where inhibition of the metastatic process would logically provide clinical benefit. For example, in patients with resectable SCLC, inhibition of pro-metastatic pathways in the neo-adjuvant and/or adjuvant setting could reduce the frequency or multiplicity of metastatic relapse.

More generally, the transition from a neuroendocrine state to a state where neuroendocrine differentiation is decreased but neuronal differentiation is increased may be related to the exceptional plasticity of SCLC cells (reviewed in *Yuan et al., 2019*). The epithelial-to-mesenchymal transition (EMT) is thought to contribute to migration, metastasis, and resistance to treatment in many cancer contexts and may play a role in SCLC (*Singh and Settleman, 2010*; *Cañadas et al., 2014*; *Krohn et al., 2014*; *Allison Stewart et al., 2017*; *O'Brien-Ball and Biddle, 2017*). Vascular mimicry (or epithelial-to-endothelial transition (EET) *Yuan et al., 2019*) may also contribute to tumor growth and response to treatment in SCLC (*Williamson et al., 2016*). Similarly, Notch-induced dedifferentiation to a non-neuroendocrine state can generate an intra-tumoral niche that protects neuroendocrine SCLC cells (*Lim et al., 2017*). Based on our results and recent observations in other cancers (*Wingrove et al., 2019*), we propose that an epithelial-to-neuronal transition contributes to key aspects of cancer metastasis. Further characterization of this neuronal state in both neuroendocrine and non-neuroendocrine cancers is likely to uncover novel mechanisms of cancer progression and may ultimately offer new insight into anti-metastasis strategies in the clinic.

## Materials and methods

### Mouse model

All experiments were performed in accordance with Stanford University Institutional Animal Care and Use Committee guidelines. *Trp53$^{flox}$*, *Rb1$^{flox}$*, *p130$^{flox}$*, and *R26$^{mTmG}$* mice have been described (*Muzumdar et al., 2007*; *Schaffer et al., 2010*; *Denny et al., 2016*) (RRID:MMRRC_043692-UCD). Tumors were initiated by inhalation of Adeno-CMV-Cre (University of Iowa Vector Core, Iowa city, Iowa) as described in *Denny et al. (2016)*, following a published protocol (*DuPage et al., 2009*).

### Cell culture

All murine and human SCLC cell lines used in this study grow as floating aggregates and were cultured in RPMI with 10% FBS, 1 × GlutaMax, and 100 U/mL penicillin-streptomycin (Gibco, Thermo Fisher Scientific, Waltham, MA). Human cell lines were originally purchased from ATCC and cell identities were validated by Genetica DNA Laboratories using STR analysis. NJH29 SCLC cells were derived from a patient-derived xenograft (PDX), which has been described (*Jahchan et al., 2013*). The LU86 and LU102 models were obtained from Stemcentrx (*Saunders et al., 2015*). The JHU-LX102 (LX102) model was a kind gift from Dr. Watkins (*Leong et al., 2014*). The murine cell lines were described (*Denny et al., 2016*; *Yang et al., 2018*). Briefly, 16T and KP22 cells are from individual primary tumors from the lungs of *Rb/p53 DKO* mice. N2N1G cells were derived from a lymph node metastasis in an *Rb/p53/p130 TKO; Rosa26$^{mTmG}$* mouse. 6PF cells were derived from metastatic cells in the plural fluid in an *Rb/p53/p130 TKO; Rosa26$^{mTmG}$* mouse. All cell lines were confirmed to be mycoplasma-negative (MycoAlert Detection Kit, Lonza, Basel, Switzerland).

### In vitro 2d matrigel migration and protrusion assay

Silicone inserts (ibidi 80209, Grafelfing, Germany) were attached to wells in 12-well (up to two inserts) or 24-well (one insert) plates pre-coated with poly-D-lysine for 15 min (Sigma-Aldrich, St. Louis, MO).~$8 \times 10^5$ cells were seeded to each chamber of the insert in 100 µL resulting in cells at ~80–90% confluency. After at least 6 hr, the inserts were carefully removed and 0.75–1 mL of a 1:1 Matrigel (Corning, Corning, NY): cell culture media mix was slowly added to cover each well. 1 mL of cell culture media was added on top of the solidified Matrigel to prevent drying. For quantification of cell migration and protrusions, the number of cells and the number of protrusions were counted in the gap at 10x under the microscope. The time points (between 36 hr and 96 hr) were dependent on the growth rate of the cell populations.

### In vitro emigration assay

1:1 Matrigel (Corning, Corning, NY):cell culture media mix containing SCLC spheres were added to Corning 12-well plates. The plates were incubated at 37°C, 5% CO2 for 48 hr. Images were obtained using a Keyence BZ-X710 microscope at 10X. Image analysis was carried out using ImageJ by measuring the area covered by cells that migrated out the aggregates/spheres. nine spheres in total were analyzed per condition in two independent experiments and the emigration efficiency was calculated by normalizing the area of emigration to the area of each sphere analyzed.

### Live imaging of cell migration and quantification

SCLC cells were plated as described in the 2D Matrigel migration assay and cultured for 24 hr before imaging. Then 10x DIC images were collected every 15 min for 25 hr using a Zeiss LSM 710 confocal microscope (Zeiss, Oberkochen, Germany) with a live imaging chamber set to 37°C, 5% CO2. To quantify the time-lapse videos, we examined nuclear movement and process length (as described in *Oudin et al., 2011*) using the FIJI software (NIH, Bethesda, USA). The position of the cell nucleus was tracked in each frame using the Manual Tracking plugin to obtain the distance migrated by the nucleus per frame and the average cell velocity over the entire video. Neuronal cell migration occurs via three steps: the cell extends a leading process, the nucleus translocates into the leading process via nucleokinesis, and the cell loses its trailing process. To quantify translocation events, we quantified the fractions of steps taken by each cell that were over 8 µm, which represents the length of one cell body and a nuclear translocation event. The process length was calculated by tracing a line

from the cell body to the tip of the leading process about 6 hr into the video. Over 30 cells were tracked and analyzed per condition.

## Immunostaining of cells in culture and human and mouse tissues

Cells were fixed with 4% PFA for 15 min, permeabilized with 0.1% Triton and stained for Tuj1 (BioLegend 801213, San Diego, CA, RRID:AB_2728521), TAU (Dako A0024, Santa Clara, CA, RRID:AB_10013724), and MAP2 (1:500, EMD Millipore AB5622, Burlington, MA, RRID:AB_91939), and with a goat anti-rabbit secondary antibody (Invitrogen, Cat # A32733, Waltham, MA, RRID:AB_2633282). Membrane GFP was stained (Abcam ab13970, Cambridge, UK, RRID:AB_300798) to mark SCLC cells and the expression of the other neuronal markers were checked using a fluorescence scope (Zeiss LSM 880). Staining was quantified by counting directly under the microscope (at 40x magnification).

For immunofluorescence, mouse brain and lungs were fixed in 4% PFA and embedded in paraffin. Tissues were stained for GAP43 (Abcam ab16053, Cambridge, UK, RRID:AB_443303) or CGRP (Sigma C7113, Darmstadt, Germany, RRID:AB_259000) to label neuroendocrine cells. For immunohistochemistry, mouse tumor samples were fixed in 4% formalin and paraffin embedded. Hematoxylin and Eosin (H and E) staining was performed using standard methods. For immunohistochemistry, we used antibodies to GFP (Abcam ab6673, RRID:AB_305643), UCHL1 (Sigma-Aldrich HPA005993, RRID:AB_1858560), Tuj1 (BioLegend 801213, RRID:AB_2728521), and TAU (Dako A0024, RRID:AB_10013724).

Tissue microarrays (LC818a, US Biomax, Rockville, MD) were stained for TAU and scored by a board-certified pathologist on a three point scale as follows: 0 = negative or weak staining of less than 10% cells, 1 = moderate intensity staining, 2 = strong intensity staining.

## Whole mount immunofluorescence staining and imaging of tumors

Detailed methods for whole mount immunofluorescence staining have been described (*Yang et al., 2018*). Subcutaneous tumors with 5–10% GFPpositive cells mixed with non-GFP labeled SCLC tumor cells were dissected and were fixed in 4% paraformaldehyde and sectioned with a vibrating blade microtome at 500 µm thickness. Tumor slices were optically cleared using the CUBIC method, comprised of a three-hour incubation at room temperature in CUBIC one reagent and long-term storage in CUBIC 2 at 4°C (*Susaki et al., 2015*). Sections were imaged using a Zeiss LSM 780 laser scanning confocal microscope.

For DiI staining and imaging, subcutaneously transplanted human SCLC xenograft were harvested after 3 weeks of growth and cut into 500mm ~ 1 cm thick slices. Tumor pieces were stained with the red fluorescent tracer DiI (D282, Thermo Fisher Scientific) in a spot-wise manner, incubated in 37°C, 5% CO2 chamber for 20 min and washed three times with PBS+10%FBS to remove excess DiI before imaging. Images were collected using a Leica SP5 scope (Leica, Buffalo Grove, IL) with a water immersion lens.

## Candidate gene knockdown

Stable knockdown of candidate genes was performed using lentiviral pLKO vectors and puromycin-resistance selection (Sigma-Aldrich). For lentivirus production, $7.5 \times 10^6$ HEK293T cells were seeded into 10 cm dishes and transfected with the vector of interest using PEI (Polysciences 23966–2, Warrington, PA) along with pCMV-VSV-G (Addgene #8454) envelope plasmid and pCMV-dR8.2 dvpr (Addgene #8455) packaging plasmid. The medium was changed 24 hr later. Supernatants were collected at 36 hr and 48 hr, passed through a 40 µm filter and applied at full concentration to target cells. Two days after transduction cells were selected with Puromycin (2 µg/mL, Thermo Fisher Scientific, Waltham, MA) for at least 1 week. Knockdown was confirmed by RT-qPCR as in *Denny et al. (2016)* and immunoblot analysis. NFIB knock-down and its validation is described in *Denny et al. (2016)*. Table S8 shows the sequences of the oligonucleotides used to knock down the candidate genes. Note that the expression of the shRNAs targeting GFP partially decreased GFP expression, but cancer cells were still GFP[positive] and could be detected by flow cytometry and microscopy.

## Immunoblot analysis

GAP43 (Abcam ab16053, Cambridge, UK, RRID:AB_443303), FEZ1 (Cell Signaling 42480, Danvers, MA, RRID:AB_2799222), and HSP90 (BD Transduction Laboratories 610418, San Jose, CA, RRID:AB_

397798) antibodies were used to confirm the knockdown of each gene at the protein level. Briefly, denatured protein samples were run on 4–12% Bis-Tris gels (NuPage, Thermo Fisher Scientific, Waltham, MA) and transferred onto PVDF membrane. Primary antibody incubations were followed by secondary HRP-conjugated anti-mouse (Santa Cruz Biotechnology sc-2005, Santa Cruz, CA, RRID: AB_631736) and anti-rabbit (Santa Cruz Biotechnology sc-2030, Santa Cruz, CA, RRID:AB_631747) antibodies and membranes were developed with the ECL2 Western Blotting Substrate (Pierce Protein Biology, Thermo Fisher Scientific).

## Transplantation assays

For long-term metastasis assays, $3 \times 10^4$ of N2N1G cells or $1 \times 10^5$ of 16 T cells were injected intravenously injected into the lateral tail vein of NOD.Cg-Prkdcscidll2rgtm1Wjl/SzJ (NSG) mice (The Jackson Laboratories, Bar Harbor, ME - Stock number 005557, RRID:IMSR_JAX:005557). Mouse livers were harvested at 4–6 weeks after injection. Tumor number was quantified by directly counting on liver surface and also quantified by counting metastasis number or areas on the H and E sections. For subcutaneous injection, $5 \times 10^4$ cells were resuspended in 100 µL PBS and mixed with 100 µL Matrigel (Corning, 356231, Corning, NY) with four injection sites per mouse. For both subcutaneous and intravenous injections, SCLC cells were transplanted into age-matched gender-matched NSG mice. For short-term tumor seeding assays, $2 \times 10^7$ of N2N1G cells or $5 \times 10^7$ of 16 T cells were transplanted intravenously into the lateral tail vein of NSG mice. N2N1G, derived from *Rb/p53/p130 TKO; Rosa26^{mTmG}* mouse, has endogenous GFP expression and 16T, derived from *Rb/p53 TKO* mouse, was stained by live cell stain CFSE (Thermo Fisher Scientific, C34554) and washed prior to intravenous injection. 2 days after transplantation, mouse livers were harvested, dissociated into a single cell suspension and analyzed by FACS to determine the percentage of GFP^positive cancer cells. FACS data were analyzed by FlowJo.

## Single-cell RNA-seq analysis

Single-cell sequencing data from normal pulmonary neuroendocrine cells and other major airway epithelial cell types previously annotated (*Ouadah et al., 2019*) were analyzed for expression of a curated list of genes. The methods for measuring expressing of each gene in transcripts per million (tpm) are previously described (*Ouadah et al., 2019*). In this report, we imported the tpm (JO_tpm-Genes_noERCCs.txt) into Seurat v2.0 to create a seurat object and normalized using standardized methods previously described (*Butler et al., 2018*). Gene expression data were represented using heatmaps.

## Pathway and process enrichment analysis

Metascape (metascape.org, RRID:SCR_016620) was used to analyze the lists of genes involved in axonogenesis and neuronal migration. Metascape integrates data from KEGG Pathway, GO Biological Processes, Reactome Gene Sets, Canonical Pathways and CORUM (*Zhou et al., 2019*). The analysis of interactions between the top 13 candidate genes was performed using STRING (string-db.org) (*Szklarczyk et al., 2019*). The analysis of dependency upon knock-down was performed using the in the Cancer Dependency Map project (depmap.org/portal/) in February 2019 with the Combined RNAi (Broad, Novartis, Marcotte) data (*Tsherniak et al., 2017*). The human RNA-seq datasets for human SCLC are from *George et al. (2015)*. Data from the Cancer Cell Line Encyclopedia (CCLE) are available at the Expression Atlas (https://www.ebi.ac.uk/gxa/home). The complete RNA-seq analysis of KP22 and N2N1G mouse cells will be published elsewhere, but the data are available in *Supplementary file 2*–table 9. The mouse primary tumors and metastases datasets are from *Denny et al. (2016)*; *Yang et al. (2018)*.

## Statistics

Statistical significance was assayed with GraphPad Prism software (RRID:SCR_002798). The statistical tests used, the numerical p-values, and the number of independent replicates is indicated in the figure legends.

## Acknowledgements

We thank Pauline Chu and Jon Mulholland for technical assistance; Alexandra Orantes and Alyssa Ray for administrative support; Kang Shen, Pengpeng Li, Gregor Bieri, Nick Kramer, Eric Gardner, Gregory Giannone, and Olivier Rossier for helpful discussions; and members of the Winslow and Sage laboratories for helpful comments. We thank Dr. Charlie Rudin and Stemcentrx for the PDX models. We thank the Stanford Shared FACS Facility and Cell Sciences Imaging Facility. This work was supported by NIH R01 CA206540 (to JS) and in part by the Stanford Cancer Institute support grant (NIH P30 CA124435). DY was supported by a Stanford Graduate Fellowship and by a TRDRP Dissertation Award (24DT-0001). FQ was supported by a Damon Runyon Postdoctoral Fellowship. HC was supported by a Tobacco-Related Disease Research Program Postdoctoral Fellowship. C-HC was supported by an American Lung Association Fellowship. BMG was supported by the Pancreatic Cancer Action Network – AACR Fellowship in memory of Samuel Stroum (14-40-25-GRUE), was a Hope Funds for Cancer Research Fellow supported by the Hope Funds for Cancer Research (HFCR-15-06-07), and is a recipient of an Emmy Noether Award from the German Research Foundation (DFG). MJO was supported by NIH R00 CA207866. JS is the Harriet and Mary Zelencik Scientist in Children's Cancer and Blood Diseases.

## Additional information

### Competing interests

Monte M Winslow: has equity in, and is an advisor for, D2GOncology. Julien Sage: Receives research funding from Stemcentrx/Abbvie, Pfizer, and Revolution Medicines and owns stock in Forty Seven Inc. The other authors declare that no competing interests exist.

### Funding

| Funder | Grant reference number | Author |
| --- | --- | --- |
| National Cancer Institute | NIH R01 CA206540 | Julien Sage |
| National Cancer Institute | P30 CA124435 | Monte M Winslow<br>Julien Sage |
| Tobacco-Related Research Program | Re-search 24DT-0001 | Dian Yang |
| Damon Runyon Cancer Research Foundation | | Fangfei Qu |
| Tobacco-Related Disease Research Program | | Hongchen Cai |
| American Lung Association | | Chen-Hua Chuang |
| Pancreatic Cancer Action Network | | Barbara M Grüner |
| Hope Funds for Cancer Research | | Barbara M Grüner |
| National Cancer Institute | R00 CA207866 | Madeleine J Oudin |

The funders had no role in study design, data collection and interpretation, or the decision to submit the work for publication.

### Author contributions

Dian Yang, Christina Kong, Conceptualization, Formal analysis, Investigation, Visualization; Fangfei Qu, Formal analysis, Investigation, Visualization; Hongchen Cai, Chen-Hua Chuang, Jing Shan Lim, Nadine Jahchan, Barbara M Grüner, Investigation; Christin S Kuo, Resources, Formal analysis, Investigation, Visualization; Madeleine J Oudin, Conceptualization, Formal analysis, Supervision, Funding acquisition, Investigation, Visualization; Monte M Winslow, Conceptualization, Supervision, Funding acquisition, Visualization; Julien Sage, Conceptualization, Resources, Formal analysis, Supervision, Funding acquisition, Visualization

## Author ORCIDs
Barbara M Grüner (ID) http://orcid.org/0000-0003-0974-4826
Julien Sage (ID) https://orcid.org/0000-0002-8928-9968

## Ethics
Animal experimentation: All experiments were performed in accordance with Stanford University Institutional Animal Care and Use Committee guidelines (protocol number 13565).

## Decision letter and Author response
Decision letter https://doi.org/10.7554/eLife.50616.sa1
Author response https://doi.org/10.7554/eLife.50616.sa2

## Additional files
### Supplementary files
• Supplementary file 1. Key Resources table.

• Supplementary file 2. This Excel file contains all the Tables associate with the manuscript. Table 1: A Summary of all the cell line that are used and tested for protrusion formation. Table 2: Expression of genes involved in axonogenesis and axon guidance in primary human SCLC tumors by RNA-seq Table3: Expression of genes involved in axonogenesis and axon guidance in mouse SCLC tumors and metastases by RNA-seq Table 4: Metascape analysis of the top 20 clusters with their representative enriched terms (one per cluster) for the 69 candidates. Table 5: Biological process GO term analysis for the 13 selected proteins Table 6: 13 candidate genes that may control the growth of axonal-like protrusions on SCLC cells Table 7: Expression levels and dependency scores for the 13 candidates in human SCLC cells Table 8: Knock-down of the 13 candidate genes in mouse SCLC cells Table 9: RNA-seq analysis of N2N1G and KP22 mouse SCLC cells

• Transparent reporting form

### Data availability
All data generated or analyzed during this study are included in the manuscript and supporting files. The RNA-seq data for primary human SCLC is available in Table S10 of George et al., 2015. The full dataset can be obtained after approval from the University of Cologne upon request with the corresponding author.

The following previously published datasets were used:

| Author(s) | Year | Dataset title | Dataset URL | Database and Identifier |
|---|---|---|---|---|
| Yang D, Greenside P, Sage J, Winslow M | 2018 | Inter-tumoral heterogeneity in SCLC is influenced by the cell-type of origin | https://www.ncbi.nlm.nih.gov/geo/query/acc.cgi?acc=GSE116977 | NCBI Gene Expression Omnibus, GSE116977 |

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
