## [Decision Letter]

**Acceptance summary:**

The paper provides strong evidence that axon-like protrusions on SCLC cells in culture and in vivo contribute to the aggressive, metastatic potential of these cells. Protrusions were detected in mouse SCLC cell lines in migration assays and in vivo, in PDX models. Axonal markers such as GAP43 are expressed in mouse SCLCs, about 50% of human primary SCLC and 6 out of 9 human SCLC metastases, but not in normal neurendocrine or non-endocrine lung epithelial cells or normal mouse lung sections. While the driver that activates axonal protein expression is not clear (e.g., expression of NEUROD1 or ASCL1 does not correlate, and NFIB appears to be neither necessary nor sufficient), the axonal proteins do seem to be important for migration and metastasis: when GAP43 or other proteins involved in axonal biology are knocked down by shRNA, cell protrusions and the migratory activity of the cells decrease. Importantly, this study also shows in an in vivo mouse model of metastasis, knocking down GAP43 or FEZ1, decreases metastasis to the liver, without affecting primary tumor formation. The concept that the neuronal characteristics of SCLC are correlated with poor outcome is in the literature, but this study provides identification of some players and experimental evidence for their requirement for migration and metastasis. Overall, these findings are significant for our understanding of SCLC migration and metastasis and may represent a more general new mechanism for cancer cells to promote their migration and metastasis.

**Decision letter after peer review:**

Thank you for submitting your article "Axon-like protrusions promote small cell lung cancer migration and metastasis" for consideration by *eLife*. Your article has been reviewed by three peer reviewers, and the evaluation has been overseen by Jonathan Cooper as the Senior and Reviewing Editor. The following individuals involved in review of your submission have agreed to reveal their identity: John Minna (Reviewer #1); René-Marc Mège (Reviewer #3).

The reviewers have discussed the reviews with one another and the Reviewing Editor has drafted this decision to help you prepare a revised submission.

Summary:

Small cell lung cancer (SCLC) is highly metastatic but there is little understanding of what features of SCLC cells contribute to their propensity to invade and metastasize. This study proposes that progressive neuronal characteristics contribute to the aggressive, metastatic potential of these cells. Yang et al. characterize axon-like protrusions that they observe on SCLC cells in culture and in vivo. The protrusions that express the axonal markers TAU and TUJ1 but not the dendritic marker MAP2. They observed protrusions in mouse SCLC cell lines in migration assays, in vivo, and in PDX models. When multiple proteins involved in axonal biology are knocked down by shRNA, cell protrusions and the migratory activity of the cells decrease. Importantly, this study also shows in an in vivo mouse model of metastasis, knocking down 2 genes, GAP43 and FEZ1, decrease metastasis to the liver, without affecting primary tumor formation. The concept that the neuronal characteristics of SCLC are correlated with poor outcome is in the literature, but this study provides identification of some players and experimental evidence for their requirement for migration and metastasis. This neuroendocrine/epithelial to neuronal transition may also be shared by other cancer cells and may thus represent a more general new mechanism for cancer cells to promote their migration and metastasis.

Essential revisions:

This paper is reviewed in the context of the urgent need to identify and potentially therapeutically target mechanisms of metastases in SCLC, one of the most highly metastatic human cancers for which we need rational therapies. Thus, the authors have identified and tackled an important problem with clinical translational relevance. There are a series of issues, which, if addressed, would improve this manuscript and strengthen the conclusions.

1) While it is clear that these SCLC-expressed neuronal genes are functionally important, the authors conclude that they have identified a cellular mechanism through which neuroendocrine to neuronal transition promotes metastasis of SCLC cells (Discussion, first paragraph). An alternative possibility is that these programs are active in normal lung neuroendocrine cells and in primary SCLC, contributing to a high propensity of SCLC cells to metastasize. What is unclear is the extent to which a neuroendocrine to neuronal transition is at work in promoting the expression of the genes under study and conferring the phenotypes observed.

For example, do normal pulmonary neuroendocrine cells (PNECs) form long protrusions/axon-like structures and express the axonal marker TAU? Do normal PNECs express GAP43 and FEZ1, i.e. the neuronal genes most thoroughly examined in this study shown by the authors to be important for metastasis?

Also, is there evidence that the expression of genes such as GAP43 and FEZ1 is increased in metastatic SCLC compared to primary SCLC? These genes were expressed in the George et al., RNAseq dataset composed of primary SCLC samples, so it is important to know if they increase in metastatic cells.

If the neuronal features of SCLC cells identified in this study are active in normal lung neuroendocrine cells and in primary SCLC, then it becomes less clear that a neuroendocrine to neuronal transition is important for SCLC metastasis and for the phenotypes described and the authors would need to reconsider their conclusions.

2) There are many human and mouse SCLC cell culture models available for study. How many were analyzed for the presence of protrusions and what% had axon like protrusions? The text is vague on this point and only few cell lines are shown. Likewise, in deposited datasets like those the authors have studied (e.g. George et al., 2015) how frequent are SCLC expressing the neuronal markers? Are there molecular, clinical, and histological differences between the SCLC preclinical models that do or don't have these protrusions?

3) The authors are very vague on the preclinical model cell line and patient derived xenograft (PDX) descriptions. Basically, readers should not have to go look up the information on them to understand the experiments. It would be really helpful if any given model used could be described better to give some context. For example, what are the characteristics of the different human PDXs used-are the expressing key factors or characteristics that have been described for different subtypes of SCLC such as ASCL1, NEUROD1, POU2F3, none of these, etc.? The authors were active in publications studying such factors and could easily provide quantitative information and context of neuronal differentiation to such factors as these lineage oncogenes (beyond the comparison of ASCL1 positive mouse models and one human SCLC line).

4) The authors were the first to report on the role of NFIB in generating metastatic behavior and indicated its role in neuronal phenotypes. In fact, in that report they raised the question of whether NFIB does this through generating neuronal characteristics. Thus, an obvious question is whether NFIB controls the expression of any of the neuronal genes they studied with functional genomics? While they cite unpublished studies of exogenous expression of NFIB, a more obvious question is whether knockout of NFIB in tumor models with protrusions influences these? I am sure, this had to be one of the first things they addressed, but we are given no information on this, making me wonder why not? Whatever the answer is we need to know.

5) Murine SCLC lines 16T and KP22 have different abilities to form protrusions (16T does and KP22 doesn't). Is there RNAseq data on these and any differentially expressed genes (DEGs) that would be of interest in this respect?

6) Assuming they are correct in the major claim (SCLCs differentiating along neuronal pathways lead to mechanisms that promote metastases), what is the path to a "therapeutic window"? Clearly, targeting some of these features directly, could lead to injury to the nervous system. So, is the approach to be to decide what controls this expression program and then target that or something else? This brings one back to the consideration of the role of key lineage oncogenes (in this case NEUROD1 seems like the prime culprit) or transcription factors like NFIB in controlling the neuronal program, and how best to target these transcription factors. This key issue is only dealt with by the authors in a tangential way in the Results or Discussion section.

7) The authors show that metastasis is affected very early (two days). This suggests that GAP43 or FEZ1 inhibition impacts the emigration out of tumor. This may be difficult to detect rapidly by in vivo imaging in mice but an easy way to start could be to seed tumor cells aggregates in 3D Matrigel or collagen gel and image live cell emigration. The authors have all the tools to test this hypothesis in a very short time period.

---

## [Author Response]

Essential revisions:This paper is reviewed in the context of the urgent need to identify and potentially therapeutically target mechanisms of metastases in SCLC, one of the most highly metastatic human cancers for which we need rational therapies. Thus, the authors have identified and tackled an important problem with clinical translational relevance. There are a series of issues, which, if addressed, would improve this manuscript and strengthen the conclusions.1) While it is clear that these SCLC-expressed neuronal genes are functionally important, the authors conclude that they have identified a cellular mechanism through which neuroendocrine to neuronal transition promotes metastasis of SCLC cells (Discussion, first paragraph). An alternative possibility is that these programs are active in normal lung neuroendocrine cells and in primary SCLC, contributing to a high propensity of SCLC cells to metastasize. What is unclear is the extent to which a neuroendocrine to neuronal transition is at work in promoting the expression of the genes under study and conferring the phenotypes observed.For example, do normal pulmonary neuroendocrine cells (PNECs) form long protrusions/axon-like structures and express the axonal marker TAU? Do normal PNECs express GAP43 and FEZ1, i.e. the neuronal genes most thoroughly examined in this study shown by the authors to be important for metastasis?

We now present extensive data on primary neuroendocrine cells, early stage primary tumors, and metastasis, all of which support a model in which these neuronal programs are specifically induced during SCLC progression. We presented data in the first version of the manuscript that TAU expression is lowly expressed or not detectable in > 90% of early SCLC primary tumors in the mouse models (Figure 4—figure supplement 1A-B). Importantly, the percentage of tumors that express TAU increased as the tumors progress. We also presented data that TAU is undetectable in ~45% of human SCLC primary tumors (Figure 2—figure supplement 2B). These data suggested that the neuronal programs involved in axonogenesis and neuronal migration are not activated or activated at low levels in early lesions. To provide some additional insight into this pattern of expression of neuronal markers in SCLC, we analyzed the expression of the axonogenesis marker GAP43 in human tissue microarrays. We found that ~half of the human primary tumors do not express GAP43, indicating that the neuronal programs are not always present in SCLC tumors. These new data are now mentioned in the text, with representative images and quantification shown in new Figure 3—figure supplement 3B. These correlative data are consistent with a model in which the axonogenesis and neuronal migration programs are turned on during tumorigenesis.

At the time of our initial submission, we had not examined the expression of neuronal markers associated with axon-like protrusions in non-transformed neuroendocrine cells. In part, this was because SCLC tumors are likely to arise from multiple different cell types in the lung, not only from neuroendocrine cells. Previous work in mouse models from the lab of Dr. Anton Berns (1) and our lab (2) suggested that neuroendocrine cells may be a cell of origin without excluding other cell types. Recently, the group of Dr. Mark Krasnow published data that SCLC tumors can arise from a small subpopulation of neuroendocrine cells (3). However, new work from the group of Dr. Anton Berns (4) and our recent work (5) support the idea that non-neuroendocrine lung epithelial cells may also serve as cell types of origin for SCLC. In particular, a recent study from the group of Dr. Mirentxu Santos shows that basal cells can also be a cell type of origin for SCLC at least in specific genetic contexts (6). Thus, to address the point raised by the reviewer in an unbiased manner, with the help of Dr. Christin Kuo (now a co-author on the manuscript), we analyzed single-cell RNA-seq data in which neuroendocrine and non-neuroendocrine lung epithelial cells were studied (3). This analysis shows that the neuronal programs associated with axonogenesis and neuronal migration are not expressed (as a whole) in any of the cell populations analyzed. These new data are now mentioned in the text and in new Figure 2—figure supplement 2C. Furthermore, we stained lung sections from adult mice for GAP43 and found no detectable expression in CGRP+ neuroendocrine cells (or in any other lung epithelial cell types) (new Figure 2—figure supplement 2D). Based on these observations, and on the difficulty of growing primary neuroendocrine cells in culture, we did not pursue studies to investigate whether primary neuroendocrine cells form protrusions.

Together, these data further support a model in which the neuronal migration and axonogenesis programs are specifically turned on during SCLC progression.

Also, is there evidence that the expression of genes such as GAP43 and FEZ1 is increased in metastatic SCLC compared to primary SCLC? These genes were expressed in the George et al., RNAseq dataset composed of primary SCLC samples, so it is important to know if they increase in metastatic cells.

From the analysis of human primary tumors (e.g. tissue microarrays), it is difficult to know if the tumors studied have already gained metastatic potential, however the observation that only a fraction of primary tumors express GAP43 is suggestive of a switch, as discussed above. Through our clinical collaborators, we obtained 9 human SCLC metastases and stained them for GAP43 expression. Six of these metastases expressed GAP43. Although these numbers are low, these new data show that GAP43 is expressed in a fraction of metastatic SCLC (new Figure 4—figure supplement 1C).

To complement the analysis of human tumors (which is limited by the paucity of metastatic samples, especially of paired primary tumor and metastasis samples from the same patients), we present RNA-seq data from primary tumors and metastases in our SCLC mouse models (Supplementary file 2—table 3). We realize that it was difficult to extract key information from the large table that we initially included in our manuscript. Thus, to address this point, we generated a new figure plotting the fold change for the 13 key neuronal genes in metastases versus primary tumors (only 12 of these genes are actually present in our mouse RNA-seq datasets). We found that the increase in the top candidate genes is significant in the mouse model in which tumors are initiated by Adeno-CMV-Cre instillation. It is also important to remember that this likely represents an underestimation of the induction of these genes because some primary tumors in this study are already in the metastatic state. These data now are shown in new Figure 4—figure supplement 1D. These data provide support to the idea that these neuronal genes are increase during metastatic progression in certain contexts.

If the neuronal features of SCLC cells identified in this study are active in normal lung neuroendocrine cells and in primary SCLC, then it becomes less clear that a neuroendocrine to neuronal transition is important for SCLC metastasis and for the phenotypes described and the authors would need to reconsider their conclusions.

As discussed above, our data strongly support a model in which the gene programs associated with neuronal migration and axonogenesis are upregulated during SCLC development.

2) There are many human and mouse SCLC cell culture models available for study. How many were analyzed for the presence of protrusions and what% had axon like protrusions? The text is vague on this point and only few cell lines are shown.

We agree that providing a more a clear description of each model is important. To address this point (and point #3 below), we generated a new table in which we compiled information related to the mouse and human models analyzed (new Supplementary file 2—table 1).

Likewise, in deposited datasets like those the authors have studied (e.g. George et al., 2015) how frequent are SCLC expressing the neuronal markers? Are there molecular, clinical, and histological differences between the SCLC preclinical models that do or don't have these protrusions?

In Figure 3—figure supplement 1A, we show the expression of the 13 top candidate genes in the George et al. human SCLC dataset. This analysis shows variable expression of this gene signature but it has not been possible to access the clinical data for each patient and correlate gene expression and clinical data. To address this point at least partly, in the revised version of the manuscript, we performed a correlation analysis between the expression of these genes and some key drivers of SCLC, such as the ASCL1 and NEUROD1 transcription factors. We found a better correlation for this gene signature and higher NEUROD1 levels, but a number of our cellular models with protrusions belong to the ASCL1-high subtype. This analysis failed to identify a clear driver of the switch to increased expression of axonogenesis and migration programs. It is possible that larger datasets would prove more informative. These data are shown in new Figure 3—figure supplement 1C.

3) The authors are very vague on the preclinical model cell line and patient derived xenograft (PDX) descriptions. Basically, readers should not have to go look up the information on them to understand the experiments. It would be really helpful if any given model used could be described better to give some context. For example, what are the characteristics of the different human PDXs used-are the expressing key factors or characteristics that have been described for different subtypes of SCLC such as ASCL1, NEUROD1, POU2F3, none of these, etc.? The authors were active in publications studying such factors and could easily provide quantitative information and context of neuronal differentiation to such factors as these lineage oncogenes (beyond the comparison of ASCL1 positive mouse models and one human SCLC line).

This is a very good point and we apologize for not making these completely clear in the first submission. This point is closely related to point #2 above. We had previously addressed some aspects of this question in the Discussion section of the initial manuscript, but we agree that it is better to provide clear information to the reader. We have now added this information in new Supplementary file 2—table 1. Unfortunately, we do not have gene expression data or copy number data for all the cellular models that we used, thus this correlative analysis remains informative but incomplete. As an additional piece of data, we examined the expression of key drivers of SCLC phenotypes in human SCLC cell lines and their possible relationship with the growth of protrusions in culture. We found no correlation between ASCL1 and NEUROD1 levels and the ability to grow protrusions (new Figure 3—figure supplement 1C-D). Thus, the progression to this cellular state appears to not be constrained within one SCLC subtypes and is detected in a subset of both mouse and human cell culture system and in vivo tumors.

4) The authors were the first to report on the role of NFIB in generating metastatic behavior and indicated its role in neuronal phenotypes. In fact, in that report they raised the question of whether NFIB does this through generating neuronal characteristics. Thus, an obvious question is whether NFIB controls the expression of any of the neuronal genes they studied with functional genomics? While they cite unpublished studies of exogenous expression of NFIB, a more obvious question is whether knockout of NFIB in tumor models with protrusions influences these? I am sure, this had to be one of the first things they addressed, but we are given no information on this, making me wonder why not? Whatever the answer is we need to know.

This is an important point that we have now clarified in the revised version of our manuscript. To provide some context, our previous work, along with work from the groups of Dr. Anton Berns and Dr. David MacPherson identified NFIB upregulation as a determinant of SCLC metastasis in mouse models. In humans, NFIB is expressed in 50-75% of advanced tumors/metastases (8-10). In our previous work, NFIB upregulation was associated with increased expression of neuronal gene expression programs.

Some of our data support a role for NFIB in controlling the gene expression programs involved in the growth of protrusions in SCLC cells. For instance, as mentioned above, we found that the top candidate genes are significant upregulated in SCLC tumors that are initiated by Adeno-CMV-Cre (the model in which NFIB is nearly always upregulated in metastases) (new Figure 4—figure supplement 1D). In human and mouse SCLC tumors, the expression of our top candidate genes correlates with NFIB expression (new Figure 3—figure supplement 1C-D). Furthermore, we found a modest correlation between NFIB levels and the ability of human SCLC cell lines to grow protrusions (new Figure 3—figure supplement 1D). Finally, knock-down of NFIB in 16T cells, which form protrusions, leads to decreased formation of protrusions (new Figure 3—figure supplement 1F).

However, our data also indicate that upregulation of NFIB is not necessary or sufficient for the induction of genes programs related to axonogenesis and migration, and for the growth of protrusions in SCLC cells. For instance, expression of individual top candidate genes does not correlate with NFIB knock-down in 16T cells (which form protrusions) or with NFIB overexpression in KP22 cells (which do not form protrusions), including the genes coding for GAP43 and FEZ1 (new Figure 3—figure supplement 1E). Importantly, ectopic expression of NFIB in KP22 cells is also not sufficient to induce the formation of protrusions (new Figure 3—figure supplement 1G).

We have now added these new data to the manuscript and updated the Discussion to indicate that NFIB may be a regulator of the gene programs studied in this manuscript, but that other factors are also likely to be involved.

5) Murine SCLC lines 16T and KP22 have different abilities to form protrusions (16T does and KP22 doesn't). Is there RNAseq data on these and any differentially expressed genes (DEGs) that would be of interest in this respect?

We have performed RNA-seq on several cell lines, including 16T, N2N1G, and KP22, however the 16T data is from an experiment performed at a different time with a different sequencing platform. Thus, we focused on the N2N1G/KP22 comparison since N2N1G also grows protrusions similar to 16T. We found that the genes associated with the growth of protrusions (as determined by the impact of gene knock-down on protrusion formation) were more highly expressed in N2N1G cells compared to KP22 cells. These data are now shown in a heat map in new Figure 3—figure supplement 1E.

6) Assuming they are correct in the major claim (SCLCs differentiating along neuronal pathways lead to mechanisms that promote metastases), what is the path to a "therapeutic window"? Clearly, targeting some of these features directly, could lead to injury to the nervous system. So, is the approach to be to decide what controls this expression program and then target that or something else? This brings one back to the consideration of the role of key lineage oncogenes (in this case NEUROD1 seems like the prime culprit) or transcription factors like NFIB in controlling the neuronal program, and how best to target these transcription factors. This key issue is only dealt with by the authors in a tangential way in the Results or Discussion section.

We had begun to discuss this in the initial version of the manuscript, but we agree that these points required additional clarification. In the correlation analyses that we performed, we identified NFIB as an important regulator of the neuronal gene programs involved in the growth of protrusions (even though NFIB up-regulation is not sufficient to recapitulate these gene programs). Thus, we envision two possible therapeutic approaches. One would be to decrease NFIB levels, as our best candidate for a regulator of the protrusions. The second one would be to target individual proteins required for the formation of protrusions. The diversity of proteins required for the formation of these protrusions may thus represent the most fruitful avenue for this type of investigation. In regards to a “therapeutic window”, it is important to note that while many of these pathways are important for neuronal pathfinding and development, they may have a much less significant role in the adult. We have updated the Discussion to include these points.

7) The authors show that metastasis is affected very early (two days). This suggests that GAP43 or FEZ1 inhibition impacts the emigration out of tumor. This may be difficult to detect rapidly by in vivo imaging in mice but an easy way to start could be to seed tumor cells aggregates in 3D Matrigel or collagen gel and image live cell emigration. The authors have all the tools to test this hypothesis in a very short time period.

The two-day assays that we describe in the manuscript are based on intravenous transplantation followed by SCLC seeding in the liver. Thus, these data actually relate to the early events of metastatic seeding, rather than cancer cells extravasation out of the tumor.

However, we do agree that experiments aimed at addressing the effect of neuronal programs on emigration out of primary tumors (like those described by the reviewer) are warranted and of potential importance. Thus, to address this point, we developed a new assay in which control and knock-down aggregates (spheroids) are plated directly into Matrigel followed by quantification of the migration of SCLC cells away from the aggregates. These new data confirm the loss of migration ability when *Gap43* and *Fez1* are knocked-down (new Figure 3—figure supplement 3F-G). These data support a model in which these genes (and likely the development of protrusion which they control) may be important at multiple steps in the metastatic cascade.

Selected References:

1) Sutherland KD, Proost N, Brouns I, Adriaensen D, Song JY, Berns A. Cell of origin of small cell lung cancer: inactivation of Trp53 and rb1 in distinct cell types of adult mouse lung. Cancer Cell. 2011;19(6):754-64. Epub 2011/06/15. doi: S1535-6108(11)00167-X [pii]

10.1016/j.ccr.2011.04.019. PubMed PMID: 21665149.

2) Park KS, Liang MC, Raiser DM, Zamponi R, Roach RR, Curtis SJ, Walton Z, Schaffer BE, Roake CM, Zmoos AF, Kriegel C, Wong KK, Sage J, Kim CF. Characterization of the cell of origin for small cell lung cancer. Cell Cycle. 2011;10(16):2806-15. Epub 2011/08/09. doi: 17012 [pii]. PubMed PMID: 21822053; PMCID: 3219544.

3) Ouadah Y, Rojas ER, Riordan DP, Capostagno S, Kuo CS, Krasnow MA. Rare Pulmonary Neuroendocrine Cells Are Stem Cells Regulated by Rb, p53, and Notch. Cell. 2019;179(2):403-16 e23. doi: 10.1016/j.cell.2019.09.010. PubMed PMID: 31585080.

4) Bottger F, Semenova EA, Song JY, Ferone G, van der Vliet J, Cozijnsen M, Bhaskaran R, Bombardelli L, Piersma SR, Pham TV, Jimenez CR, Berns A. Tumor Heterogeneity Underlies Differential Cisplatin Sensitivity in Mouse Models of Small-Cell Lung Cancer. Cell reports. 2019;27(11):3345-58 e4. doi: 10.1016/j.celrep.2019.05.057. PubMed PMID: 31189116; PMCID: PMC6581744.

5) Yang D, Denny SK, Greenside PG, Chaikovsky AC, Brady JJ, Ouadah Y, Granja JM, Jahchan NS, Lim JS, Kwok S, Kong CS, Berghoff AS, Schmitt A, Reinhardt HC, Park KS, Preusser M, Kundaje A, Greenleaf WJ, Sage J, Winslow MM. Intertumoral Heterogeneity in SCLC Is Influenced by the Cell Type of Origin. Cancer Discov. 2018. doi: 10.1158/2159-8290.CD-17-0987. PubMed PMID: 30228179.

6) Lazaro S, Perez-Crespo M, Lorz C, Bernardini A, Oteo M, Enguita AB, Romero E, Hernandez P, Tomas L, Morcillo MA, Paramio JM, Santos M. Differential development of large-cell neuroendocrine or small-cell lung carcinoma upon inactivation of 4 tumor suppressor genes. Proc Natl Acad Sci U S A. 2019. doi: 10.1073/pnas.1821745116. PubMed PMID: 31611390.

7) Kuo CS, Krasnow MA. Formation of a Neurosensory Organ by Epithelial Cell Slithering. Cell. 2015;163(2):394-405. doi: 10.1016/j.cell.2015.09.021. PubMed PMID: 26435104; PMCID: PMC4597318.

8) Wu N, Jia D, Ibrahim AH, Bachurski CJ, Gronostajski RM, MacPherson D. NFIB overexpression cooperates with Rb/p53 deletion to promote small cell lung cancer. Oncotarget. 2016;7(36):57514-24.

9) Semenova EA, Kwon MC, Monkhorst K, Song JY, Bhaskaran R, Krijgsman O, Kuilman T, Peters D, Buikhuisen WA, Smit EF, Pritchard C, Cozijnsen M, van der Vliet J, Zevenhoven J, Lambooij JP, Proost N, van Montfort E, Velds A, Huijbers IJ, Berns A. Transcription Factor NFIB Is a Driver of Small Cell Lung Cancer Progression in Mice and Marks Metastatic Disease in Patients. Cell reports. 2016;16(3):631-43. doi: 10.1016/j.celrep.2016.06.020. PubMed PMID: 27373156; PMCID: PMC4956617.

10) Denny SK, Yang D, Chuang CH, Brady JJ, Lim JS, Gruner BM, Chiou SH, Schep AN, Baral J, Hamard C, Antoine M, Wislez M, Kong CS, Connolly AJ, Park KS, Sage J, Greenleaf WJ, Winslow MM. Nfib Promotes Metastasis through a Widespread Increase in Chromatin Accessibility. Cell. 2016;166(2):328-42. doi: 10.1016/j.cell.2016.05.052. PubMed PMID: 27374332.